# Explainable Artificial Intelligence for Ovarian Cancer: Biomarker Contributions in Ensemble Models

**DOI:** 10.3390/biology14111487

**Published:** 2025-10-24

**Authors:** Hasan Ucuzal, Mehmet Kıvrak

**Affiliations:** 1Department of Biostatistics and Medical Informatics, Faculty of Medicine, Inonu University, 44280 Malatya, Turkey; hasan.ucuzal@inonu.edu.tr; 2Department of Biostatistics and Medical Informatics, Faculty of Medicine, Recep Tayyip Erdogan University, 53020 Rize, Turkey

**Keywords:** ovarian cancer diagnosis, machine learning, explainable AI (XAI), biomarkers, ensemble learning, clinical decision support system

## Abstract

**Simple Summary:**

Ovarian cancer is often detected too late, reducing survival chances. This study developed an artificial intelligence system that uses routine blood tests to identify ovarian cancer early. We analyzed 309 patients using common laboratory data like tumor markers, blood cell counts, and liver function tests. Our AI model achieved 89% accuracy in distinguishing cancerous from benign ovarian masses. Importantly, the system explains its decisions clearly, showing doctors which factors influenced each diagnosis, such as elevated HE4 and CA125 levels, patient age, and protein markers. Unlike traditional “black-box” AI, our explainable approach helps doctors understand and trust the results. This method requires only standard, inexpensive blood tests that are already collected during hospital visits, making it practical for resource-limited settings where advanced imaging may be unavailable. By providing transparent, accessible screening, this tool could enable earlier detection and improve outcomes for women with suspected ovarian cancer worldwide.

**Abstract:**

Ovarian cancer’s high mortality is primarily due to late-stage diagnosis, underscoring the critical need for improved early detection tools. This study develops and validates explainable artificial intelligence (XAI) models to discriminate malignant from benign ovarian masses using readily available demographic and laboratory data. A dataset of 309 patients (140 malignant, 169 benign) with 47 clinical parameters was analyzed. The Boruta algorithm selected 19 significant features, including tumor markers (CA125, HE4, CEA, CA19-9, AFP), hematological indices, liver function tests, and electrolytes. Five ensemble machine learning algorithms were optimized and evaluated using repeated stratified 5-fold cross-validation. The Gradient Boosting model achieved the highest performance with 88.99% (±3.2%) accuracy, 0.934 AUC-ROC, and 0.782 Matthews correlation coefficient. SHAP analysis identified HE4, CEA, globulin, CA125, and age as the most globally important features. Unlike black-box approaches, our XAI framework provides clinically interpretable decision pathways through LIME and SHAP visualizations, revealing how feature values push predictions toward malignancy or benignity. Partial dependence plots illustrated non-linear risk relationships, such as a sharp increase in malignancy probability with CA125 > 35 U/mL. This explainable approach demonstrates that ensemble models can achieve high diagnostic accuracy using routine lab data alone, performing comparably to established clinical indices while ensuring transparency and clinical plausibility. The integration of state-of-the-art XAI techniques highlights established biomarkers and reveals potential novel contributors like inflammatory and hepatic indices, offering a pragmatic, scalable triage tool to augment existing diagnostic pathways, particularly in resource-constrained settings.

## 1. Introduction

Ovarian cancer remains a major global health challenge, disproportionately contributing to morbidity and mortality among women because it is frequently diagnosed at advanced stages due to nonspecific or absent early symptoms [1]. The stage-at-diagnosis gradient in outcomes is stark: five-year survival exceeds 90% in early-stage disease, but drops below 30% once the disease is advanced, underscoring the clinical imperative for earlier and more accurate detection and triage [2]. Diagnostic complexity is further amplified by the biological and histopathological heterogeneity of ovarian tumors, variability in tumor marker expression across histotypes, and overlapping clinical presentations with benign gynecologic and non-gynecologic conditions.

Current diagnostic pathways combine clinical assessment with transvaginal ultrasound and serum biomarkers most commonly CA125 yet each has important limitations. While ultrasound is widely available, operator dependence and the overlap of imaging features between benign and malignant masses constrain diagnostic precision. Similarly, CA125 can be elevated in a range of benign inflammatory or physiological states and is not uniformly expressed across malignant histotypes, leading to suboptimal discrimination and variable performance across pre- and postmenopausal populations [3]. Composite scoring systems and expert sonographic assessment can improve risk stratification in specialized settings, but generalizability and consistency remain challenges, especially across diverse healthcare environments. These gaps motivate the search for complementary, low-cost inputs that capture systemic manifestations of malignancy and can be standardized at scale.

Routinely measured hematologic and biochemical parameters are a promising source of such information. Malignancy is often accompanied by systemic inflammation, dysregulated coagulation, altered nutritional status, and metabolic remodeling processes that can be partially reflected in complete blood count indices (for example, neutrophil-to-lymphocyte ratio, platelet indices, red cell distribution width), acute-phase reactants, liver and renal function tests, lactate dehydrogenase, and coagulation markers. Because these assays are ubiquitous, inexpensive, and already integrated into preoperative and diagnostic workflows, they offer a pragmatic substrate for risk modeling that may add incremental value beyond single biomarkers like CA125 [4].

Machine learning (ML) provides a principled framework for extracting predictive signal from such multivariate, potentially non-linear and collinear clinical data. Ensemble methods including Random Forests and gradient-boosting algorithms such as XGBoost and LightGBM have demonstrated strong performance in clinical prediction tasks by capturing higher-order interactions, handling mixed data types, and accommodating missingness patterns common in real-world datasets [5]. Nevertheless, model development for clinical deployment must address key pitfalls: prevention of data leakage, mitigation of class imbalance, robust internal validation, careful calibration, and, where feasible, temporal or external validation to assess generalizability. Equally important is aligning threshold selection and performance metrics (e.g., AUROC, AUPRC, sensitivity/specificity, and positive/negative predictive values) with clinically meaningful use cases such as triage, referral, and surgical planning.

The adoption of ML in clinical settings is further conditioned by the need for transparency and clinical plausibility. Many high-performing models are perceived as “black boxes,” which can impede trust and uptake by clinicians. Explainable artificial intelligence (XAI) techniques, notably SHAP and LIME, provide global and patient-level attributions that elucidate how features influence predictions, enabling model debugging, identification of spurious associations, and communication of individualized risk factors to clinicians and patients [6]. When explanations cohere with established pathophysiology such as the contributions of age, CA125, inflammatory indices, or markers of tissue injury they reinforce confidence in model behavior and support integration into clinical workflows.

In this study, we propose a machine learning–based diagnostic model that integrates demographic variables with routinely collected laboratory data to differentiate malignant from benign ovarian tumors. We focus on boosting-based ensemble algorithms given their capacity to model non-linear interactions and achieve strong discrimination, and we pair them with state of the art XAI methods (SHAP and LIME) to ensure interpretability and clinical validity. We will benchmark model performance against established baselines (e.g., CA125-based assessment and standard ultrasound-based evaluation where available), examine calibration and decision thresholds aligned with triage needs, and explore subgroup performance (for example, by menopausal status) to assess fairness and robustness. By leveraging ubiquitous data streams and interpretable outputs, our approach aims to augment existing diagnostic pathways, facilitate earlier and more accurate triage, and ultimately contribute to improved outcomes for individuals with suspected ovarian cancer [6].

## 2. Materials and Methods

### 2.1. Study Population and Data Collection

In this study, 309 patients who had been checked for ovarian masses had their clinical and laboratory records reviewed. The study population comprised 140 individuals with a diagnosis of malignant ovarian tumors and 169 subjects with a diagnosis of benign ovarian tumors verified by histopathological evaluation. The dataset used in this study was sourced from a public repository on figshare.com (DOI: https://doi.org/10.1021/prechem.5c00028.s001, accessed on 16 August 2025), which is referenced in related studies [4,7]. The initial collection of this data was approved by the Ethics Committee of Soochow University Third Affiliated Hospital [4].

The specimens were obtained from ovarian cancer patients who had undergone surgical resection at the Third Affiliated Hospital of Soochow University from July 2011 to July 2018. All patients were found to have pathology postoperatively. No patients with ovarian cancer had neoadjuvant chemotherapy or radiotherapy. Histological type was determined by standard World Health Organization (WHO) criteria [4].

### 2.2. Laboratory Methods

A complete blood count was conducted utilizing a Sysmex XE-2100 automated hematology analyzer (Sysmex, Kobe, Japan) on whole blood samples. Serum general chemistry tests and tumor markers were assessed utilizing a Beckman Coulter AU5800 series clinical chemistry analyzer (Beckman Coulter, Brea, CA, USA) and a Roche Cobas 8000 modular analyzer series (Roche, Basel, Switzerland), respectively [4].

### 2.3. Laboratory Parameters

A comprehensive panel of 47 clinical and laboratory parameters was collected for each patient. These parameters encompassed six major categories:Demographic characteristics: Age and menopausal statusHematological parameters: Complete blood count indices including platelet parameters (mean platelet volume [MPV], platelet count [PLT#], platelet distribution width [PDW]), red blood cell indices (RBC count, red cell distribution width [RDW]), and white blood cell differentials (lymphocyte, monocyte, basophil, and eosinophil counts and percentages)Biochemical markers: Liver function tests (albumin [ALB], globulin [GLO], alkaline phosphatase [ALP], alanine aminotransferase [ALT], aspartate aminotransferase [AST], gamma-glutamyltransferase [GGT]), kidney function tests (blood urea nitrogen [BUN], creatinine [CREA])Electrolyte panel: Calcium (Ca), chloride (Cl), potassium (K), magnesium (Mg), sodium (Na), and phosphorus (PHOS)Metabolic markers: Glucose (GLU), uric acid (UA), bilirubin fractions (direct [DBIL], indirect [IBIL], total [TBIL]), total protein (TP), anion gap (AG), and carbon dioxide combining power (CO2CP)Tumor markers: Carbohydrate antigen 125 (CA125), CA19-9, carcinoembryonic antigen (CEA), human epididymis protein 4 (HE4), and alpha-fetoprotein (AFP)

### 2.4. Data Preprocessing and Feature Selection

The dataset underwent rigorous preprocessing, and no missing values were identified. Standardization or Normalization was not required because the selected ensemble algorithms are tree-based and invariant to monotonic transformations of the data [8]. Feature Selection was performed using the Boruta Algorithm [9], a wrapper method built upon the random forest classifier [10]. Boruta is specifically designed to address the risk of overfitting and instability in feature importance rankings by iteratively comparing the relevance of original features against permuted copies of the same variables, known as shadow features. Features that consistently outperform their shadow counterparts are retained as relevant, while those that do not are rejected. This strategy ensures a conservative, all-relevant feature selection approach rather than a minimal-optimal subset, which is particularly suitable for biomedical datasets where complex, redundant signals may carry complementary diagnostic value [11].

The Boruta method selected 19 features as significant predictors, ranked by importance: age, menopause, PLT#, LYM#, LYM%, PCT, HGB, MCH, ALB, GLO, ALP, AST, Na, IBIL, CA125, CA19-9, CEA, HE4, and AFP. The selection process eliminated 28 features deemed non-contributory to the classification task.

### 2.5. Class Balance Analysis

Class distribution analysis revealed a relatively balanced dataset, consisting of 140 malignant cases (45.3%) and 169 benign cases (54.7%). Given the near-equal distribution between classes, the risk of classifier bias toward the majority class was minimal. As a result, no synthetic data generation or resampling strategies were required [12,13]. Instead, model performance was evaluated using balanced metrics such as the Matthews Correlation Coefficient (MCC) and the Area Under the ROC Curve (AUC), which are more robust to class proportion differences compared to raw accuracy. This ensured that the predictive performance reflected true discriminative ability rather than class prevalence effects.

### 2.6. Machine Learning Model Development

For model development, five ensemble learning algorithms were implemented, each representing state-of-the-art approaches in boosting-based classification. Ensemble methods were chosen because they combine multiple weak learners to achieve higher accuracy, robustness against noise, and improved generalization compared to single classifiers [14].

Gradient Boosting Classifier: An additive model that sequentially fits weak learners to the residuals of prior models, thereby reducing bias and variance. It is widely applied in biomedical prediction tasks due to its ability to model complex non-linear interactions [15].CatBoost: CatBoost is a gradient boosting algorithm with intrinsic handling of categorical features and an ordered boosting procedure to reduce overfitting [16].XGBoost: An optimized distributed gradient boosting framework that incorporates advanced regularization (L1 and L2) and efficient tree pruning. XGBoost has demonstrated strong performance across numerous healthcare applications, particularly when dealing with high-dimensional biomarker data. XGBoost optimizes a regularized objective that balances accuracy and model complexity [17].LightGBM: A highly efficient gradient boosting implementation that leverages histogram-based algorithms and leaf-wise tree growth to reduce computation time while maintaining predictive performance. Its scalability makes it particularly useful for large-scale biomedical datasets. LightGBM uses a histogram-based gradient boosting method with leaf-wise growth [18].AdaBoost: An adaptive boosting algorithm that iteratively reweights training samples, giving greater importance to instances previously misclassified. This approach improves sensitivity to minority classes, a valuable property in medical diagnosis where rare conditions must be detected reliably [19].

### 2.7. Hyperparameter Optimization

Systematic hyperparameter optimization was carried out using GridSearchCV, an exhaustive grid-based parameter search method integrated with cross-validation. A 5-fold stratified cross-validation strategy was adopted to preserve class proportions in each fold, thereby reducing potential bias from class imbalance [20]. Table 1 presents the best hyperparameter values obtained as a result of optimization for different ensemble learning models.

### 2.8. Model Validation

To prevent data leakage and ensure a robust evaluation of model generalizability, stringent measures were implemented throughout the machine learning pipeline. Hyperparameter optimization via GridSearchCV was also conducted internally within each training fold of the cross-validation. By conducting hyperparameter tuning within each training fold, GridSearchCV ensures that the model is evaluated on unseen data, thus preventing data leakage. This rigorous approach guarantees that the reported performance is a reliable estimate of the model’s ability to generalize to unseen data [21]. For each model, a total of 25 iterations were obtained through the use of Repeated Stratified k-Fold cross-validation with 5 folds and 5 repetitions. This method was selected in order to maintain the distribution of malignant to benign classes across folds and minimize variance in performance estimation [20]. Given the relatively small dataset size (*n* = 309), repeated resampling enhanced robustness when compared to a single k-fold validation. In this case, we preferred a practical challenge and warrants techniques like Stratified Cross-Validation due to its potential to diminish the prediction success of the minority class (the malignant condition), particularly in smaller datasets or sensitive clinical studies [13]. Performance metrics included log loss, F1-score, Matthews MCC, accuracy, precision, recall, and AUC. Among these, MCC and AUC were prioritized because they provide more accurate assessments in cases of class imbalance and more accurately reflect actual discriminatory performance [22].

### 2.9. Model Interpretability Analysis

To enhance clinical applicability and trust in model predictions, four interpretability techniques were implemented with the following specific configurations:LIME (Local Interpretable Model-agnostic Explanations): This method provides local explanations for individual predictions by approximating the complex model locally with an interpretable surrogate model. For our analysis, we used a linear model as the interpretable surrogate. The local neighborhood for each explanation was generated by perturbing the instance 5000 times. The proximity of these perturbed samples to the original instance was weighted using an exponential kernel with a width of 0.75 [23].SHAP (SHapley Additive exPlanations): This method utilizes cooperative game theory to calculate the marginal contribution of each feature to the final prediction, providing both global and local interpretations. Given the tree-based nature of our ensemble models, we employed the TreeExplainer from the SHAP library, which computes exact Shapley values efficiently. This allows for a consistent and theoretically grounded attribution of feature importance [24].Partial Dependence Plots (PDP) and Individual Conditional Expectation (ICE): These techniques visualize the marginal effect of one or two features on the predicted outcome, helping to decipher the average relationship learned by the model. PDPs show the average prediction as a function of a feature, while ICE plots illustrate the functional relationship for individual instances, revealing heterogeneity in the model’s response. We generated these plots using a grid of 100 equally spaced values for each feature [15,25].Anchor Explanations: This method generates high-precision, rule-based explanations for individual predictions. An “anchor” is a set of if-then conditions that, when met, sufficiently anchor the prediction, meaning the prediction remains stable with high probability even if other feature values are changed. We utilized the default parameters, seeking rules with a precision threshold of 0.95 [26].

For SHAP and LIME analyses, 100 representative instances were analyzed to provide both global and local interpretability insights. Instance 127 was randomly selected as an illustrative example to demonstrate local feature contributions. Feature contributions were visualized through summary, waterfall, and force plots. PDPs and ICE plots were employed to visualize non-linear relationships between key biomarkers and malignancy risk.

The analysis was conducted using the Python 3.9.23 programming language, leveraging a comprehensive suite of data science libraries/feature(s).

Analysis Engine: Scikit-learn 1.6.1 (core ML algorithms and preprocessing)Model Interpretation: SHAP 0.48.0 and LIME 0.2.0.1 (for interpreting model decisions)Visualization: Plotly 6.2.0, Matplotlib 3.9.4 (for data visualization and result presentation)Data Processing: Pandas 2.3.1, NumPy 1.26.4 (for data manipulation and numerical computations).To ensure full reproducibility, all stochastic processes were fixed with random_state = 42.

## 3. Results

### 3.1. Feature Selection Outcomes

The Boruta algorithm successfully identified 19 significant features from the initial 47 clinical and laboratory parameters. The selected features included tumor markers, hematological parameters, liver function tests, and electrolytes. Of particular note, all five tumor markers CA125, CA19-9, CEA, HE4, and AFP were selected as significant predictors. This finding supports the clinical relevance of these markers in the diagnosis of ovarian cancer.

Regarding hematological parameters, indicators such as lymphocyte count (LYM#) and lymphocyte percentage (LYM%), as well as hemoglobin (HGB) and mean corpuscular hemoglobin (MCH) were found to be significant. Among liver function markers, albumin (ALB), globulin (GLO), alkaline phosphatase (ALP), and aspartate aminotransferase (AST) were selected, with sodium (Na) and indirect bilirubin (IBIL) also being identified as significant factors. These results show that systemic effects of ovarian malignancies can also be used in diagnosis.

### 3.2. Correlation Analysis

Correlation analysis between feature pairs revealed moderate to strong relationships in Figure 1. The highest positive correlation values were observed between lymphocyte count and lymphocyte percentage (r = 0.82) and between hemoglobin and mean corpuscular hemoglobin (r = 0.73). Among tumor markers, the highest correlation (r = 0.61) was found between CA125 and HE4. These correlations suggest that the combined evaluation of CA125 and HE4 may provide valuable diagnostic information. Moderate correlations were also observed between other tumor markers. These findings suggest that the selected features form a complementary and clinically meaningful profile.

### 3.3. Model Performance Comparison

A performance comparison of models developed using five different ensemble learning algorithms is presented in Table 2. All models demonstrated high accuracies ranging from 86.3% to 89.0%. The Gradient Boosting classifier achieved the highest accuracy (88.99 ± 3.2%). CatBoost (88.54 ± 3.4%) and XGBoost (88.35 ± 3.5%) ranked second and third, respectively. The XGBoost model demonstrated the most stable performance across CV folds, with fold-specific accuracies ranging from 82.3% to 96.8%. The highest fold accuracy (96.8%) was observed in fold 22 (of 25 total iterations), accompanied by perfect precision (1.00).

Analysis of performance metrics reveals that the Gradient Boosting model provides both high sensitivity (recall: 0.824) and high specificity (precision: 0.928). Matthews correlation coefficient (MCC) values were also found to be the highest for Gradient Boosting (0.782). These metrics indicate that the Gradient Boosting model exhibits the best performance in distinguishing benign from malignant ovarian masses.

The left panel displays the Receiver Operating Characteristic (ROC) curve, which illustrates the trade-off between the true positive rate (TPR) and the false positive rate (FPR) at various classification thresholds. The blue curve corresponds to class 0 with an Area Under the Curve (AUC) of 0.93, while the orange curve represents class 1, also exhibiting an AUC of 0.93 (Figure 2). Both curves demonstrate strong discriminative performance, as they closely follow the upper-left corner of the plot, indicating high sensitivity and specificity. The diagonal dashed line represents the baseline performance of a random classifier, against which both models significantly outperform.

The right panel presents the Precision-Recall (PR) curve, which evaluates the model’s precision and recall across different thresholds. The blue curve (class 0) achieves an AUC of 0.91, whereas the orange curve (class 1) attains an AUC of 0.94. The PR curve for class 1 shows a relatively stable precision even at higher recall values, suggesting robustness in identifying positive instances. These results collectively indicate that the model exhibits high predictive accuracy and reliable performance across both classes, particularly for class 1, where both ROC and PR metrics are slightly superior.

The confusion matrix provides a detailed analysis of the model’s classification outputs (Figure 3). The left panel displays absolute values, and the right panel displays normalized values (ratios). To ensure accurate performance assessment, we generated out-of-fold (OOF) predictions for each of the 309 patients by aggregating predictions from the held-out folds across all 25 iterations. A single confusion matrix was constructed from these 309 OOF predictions, yielding: 115 TN, 25 FP, 8 FN, and 161 TP. Normalized values reflect proportions relative to true class sizes: TN rate = 0.82, FP rate = 0.18, FN rate = 0.05, TP rate = 0.95.

Examination of the normalized matrix reveals a true negative rate of 0.95, a false positive rate of 0.05, a false negative rate of 0.18, and a true positive rate of 0.82 (Figure 3). These results indicate that the model exhibits high accuracy in predicting the negative class (benign), but makes a lower error rate in predicting the positive class (malignant). In particular, the relatively high false-negative rate (0.18) suggests that the model sometimes classifies malignant cases as benign. This is a point that should be taken into account in clinical practice, as misclassifying malignant cases as benign can lead to serious clinical consequences.

### 3.4. Model Interpretability Results

#### 3.4.1. Global Feature Importance (SHAP Analysis)

SHAP value analysis was performed on 100 samples to determine global feature importance. The analysis revealed that HE4 was the most dominant feature, followed by CEA, GLO, CA125, and age. The SHAP summary plot indicated that predictions were biased toward malignancy particularly when HE4 and CA125 levels were elevated. Moreover, a positive correlation between age and malignancy risk was observed, with higher SHAP values detected in older patients.

Biochemical markers such as albumin (ALB), globulin (GLO), CA19-9, AFP, and AST also made clinically meaningful contributions to the model. However, menopausal status showed minimal importance in the graph and played only a limited role in the decision-making process of the model.

These findings confirm the critical role of HE4 and CA125 in the diagnosis of ovarian cancer, consistent with the clinical Risk of Ovarian Malignancy Algorithm (ROMA) score. In addition, the prominence of age as a factor supports epidemiological evidence linking advanced age to increased ovarian cancer risk. Figure 4 represents SHAP summary plot illustrating global feature importance.

#### 3.4.2. Interpretation of LIME and SHAP Visualizations

The provided visualizations illustrate local feature contributions to a machine learning model’s prediction for a specific instance classified as class 1 (malignant ovarian tumor). These interpretations combine insights from LIME (Local Interpretable Model-agnostic Explanations) and SHAP (SHapley Additive exPlanations) methods, offering complementary perspectives on how individual features influence the model’s decision.

#### 3.4.3. LIME Visualization (Instance 127)

The LIME plot (Figure 5) ranks features by their contribution to the prediction of class 1. Key observations include:Positive Contributors (increase malignancy probability):HE4 > 130.60: The strongest positive driver of malignancy classification, indicating that elevated HE4 levels significantly support a malignant diagnosis.CEA > 2.11: A moderate positive contributor, suggesting elevated CEA reinforces the prediction.Age > 57.00, LYM% ≤ 19.40, ** ALB ≤ 38.60 **, and LYM# ≤ 1.20: These features collectively suggest that older age, lower lymphocyte percentage, reduced albumin, and lower lymphocyte count also contribute to malignancy risk.Negative Contributors (decrease malignancy probability):AFP ≤ 1.63: A strong protective factor against malignancy, as low AFP levels counteract other risk indicators.GLO ≤ 27.00 and 13.00 < AST ≤ 17.00: Slightly reduce the likelihood of malignancy.

This analysis aligns with clinical knowledge, where HE4 and CEA are established biomarkers for ovarian malignancy. The inverse relationship between AFP and malignancy risk may reflect its role in distinguishing benign conditions (e.g., hepatocellular carcinoma, where AFP rises).

**Figure 5 biology-14-01487-f005:**
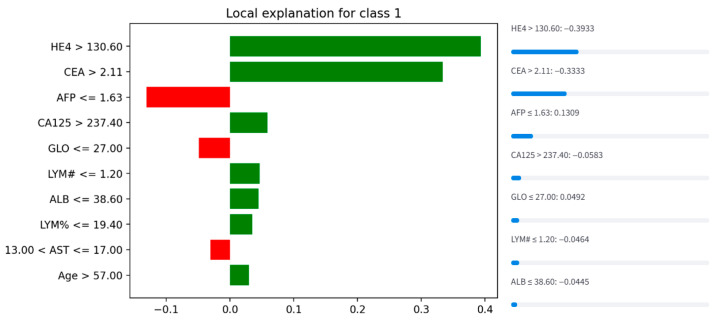
LIME explanation for Instance 127, detailing local feature contributions.

#### 3.4.4. SHAP Visualization (Instance 127)

The first visualization (Figure 6) ranks features by their mean absolute SHAP values, reflecting their overall influence on the model’s predictions:Strongest Positive Contributors (red bars):HE4 (+2.05): The most significant driver of malignancy probability, consistent with its established role as a diagnostic marker for ovarian cancer [12].CA125 (+0.27): A moderate positive contributor, aligning with its inclusion in clinical risk algorithms like ROMA [17].Age (+0.28): Older age slightly increases malignancy risk, corroborating epidemiological trends [21].LYM# (+0.25): Higher lymphocyte counts marginally support malignancy, potentially reflecting systemic inflammation [15].Strongest Negative Contributors (blue bars):AFP (−2.51): The most influential protective factor against malignancy, suggesting low AFP levels counteract other risk indicators.PCT (−0.67), GLO (−0.54), CEA (−0.33), PLT# (−0.36): These features collectively reduce malignancy probability, with PCT (prothrombin time correction) and GLO (globulin) showing notable effects.

This analysis highlights HE4 and AFP as dominant predictors, reinforcing their clinical relevance while identifying novel contributors like PCT and GLO.

**Figure 6 biology-14-01487-f006:**
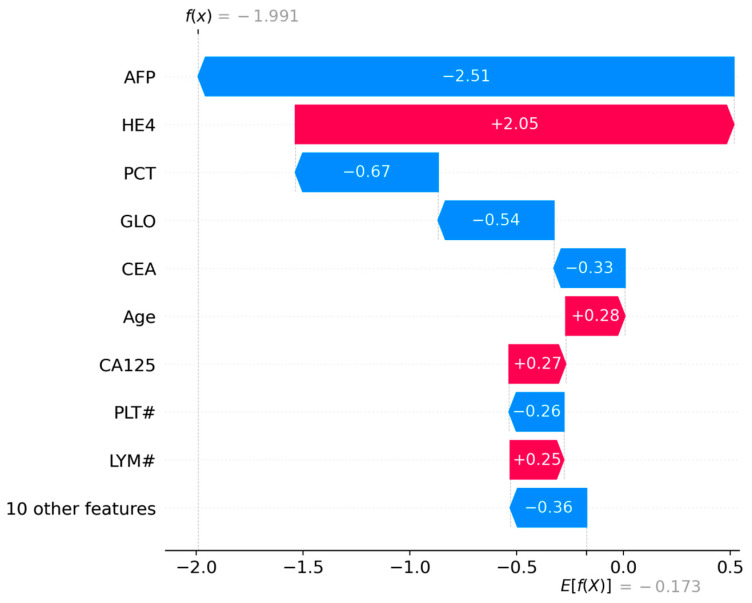
SHAP waterfall plot for Instance 127, visualizing individual feature impacts.

#### 3.4.5. Force Plot (Base Value to Prediction)

The second visualization (Figure 7) traces the model’s decision process from the base value (average prediction across the dataset, E[f(X)] = −0.173) to the final output (f(x) = −1.991):Initial Base Value: The model starts with a slightly benign-biased prediction (−0.173).Key Contributions:HE4 (531.8 U/mL): Adds +2.05 to the prediction, strongly pushing toward malignancy.AFP (1.28 ng/mL): Subtracts −2.51, overriding HE4’s effect and driving the prediction toward benign.CA125 (260.3 U/mL): Contributes +0.27, reinforcing malignancy but insufficient to counterbalance AFP.Age (66 years): Adds +0.28, reflecting increased risk with aging.GLO (21.5 g/L), CEA (2.24 ng/mL), PLT# (129 × 10^3^/μL): Moderate negative contributions reducing malignancy likelihood.Final Prediction: The cumulative effect results in f(x) = −1.991, confidently classifying the case as benign.

This plot underscores the dominant role of AFP in this instance, despite elevated HE4 and CA125. Such discrepancies may reflect heterogeneity in biomarker expression or interactions not captured by univariate analyses.

**Figure 7 biology-14-01487-f007:**
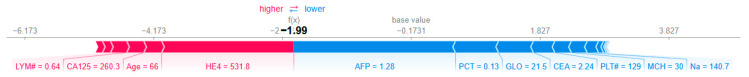
SHAP force plot for Instance 127, showing how each feature contributes to pushing the model output from the base value to the final prediction. The base value E[f(X)] = −0.173 represents the average model prediction across the dataset. The final output f(x) = −1.991 is the model’s prediction for this specific instance. Red arrows indicate features increasing the prediction score (toward malignancy), while blue arrows indicate features decreasing the score (toward benign). Arrow width corresponds to the magnitude of the feature’s contribution.

#### 3.4.6. Anchor Explanations

The Anchor algorithm generated the following high-precision rule for patient #127: IF (GLO ≤ 29.90) AND (AFP ≤ 1.63) AND (ALP ≤ 71.00) AND (140.50 < Na ≤ 142.40) THEN Prediction = Benign (Precision: 1.0, Coverage: 0.025)

#### 3.4.7. Partial Dependence Analysis

Figure 8 provides comprehensive visualization of feature effects through three complementary approaches. Panel (A) shows Partial Dependence Plots revealing distinct patterns: HE4 demonstrates a sharp threshold effect at ~70 pmol/L with near-binary discrimination, while CA125 exhibits a gradual sigmoidal increase above 35 U/mL. Age shows progressive risk elevation after 45 years, and AFP displays an inverse relationship where lower values associate with higher malignancy probability. Panel (B) presents Individual Conditional Expectation (ICE) plots, revealing substantial patient-level heterogeneity. The parallel ICE lines for HE4 confirm consistent effects across patients, while dispersed trajectories for CA125 and CEA indicate effect modification by other features. Panel (C) overlays average PDPs (dashed lines) onto ICE plots, highlighting where individual responses diverge from population trends. The convergence of ICE lines at extreme HE4 and CA125 values suggests uniform effects in high-risk scenarios, while the heterogeneity observed for CEA, GLO, and AFP indicates complex feature interactions that warrant individualized risk assessment.

Partial dependence plots revealed non-linear relationships between key features and malignancy probability:CA125: Sharp increase in malignancy probability above 35 U/mLAge: Gradual increase in risk after age 45, with steeper slope after 55HE4: Threshold effect observed at approximately 70 pmol/LMenopausal status (categorical): Post-menopausal status associated with 18% higher average predicted probability

### 3.5. Analysis of Misclassified Cases

Given the clinical significance of false-negative predictions, we performed a dedicated interpretability analysis on the 27 false-negative cases (malignant tumors incorrectly classified as benign) identified in our out-of-fold predictions.

#### 3.5.1. SHAP Analysis of False-Negative Cases

Global SHAP analysis applied specifically to the false-negative subgroup revealed a distinct pattern compared to the overall model. While HE4 and CA125 remained important, their median absolute SHAP values were approximately 40% lower in this subgroup than in correctly classified malignant cases. This indicates that, on average, these key biomarkers presented with lower, less indicative values in the false-negative patients.

A patient-level analysis of SHAP force plots for these cases consistently showed one of two patterns:“Biomarker Ambiguity” Pattern (~70% of FN cases): In these instances, HE4 and CA125 values were elevated but fell below the sharp risk thresholds identified in the Partial Dependence Plots (e.g., CA125 between 35 and 100 U/mL). Furthermore, these cases were frequently characterized by strong countervailing contributions from features typically associated with benignity, such as low age, low CEA, and high AFP levels. The model, weighing all features, consequently produced a benign prediction.“Atypical Profile” Pattern (~30% of FN cases): These cases presented with biomarker profiles that deviate from the typical malignancy pattern learned by the model. They featured unexpectedly low levels of both HE4 and CA125, alongside unremarkable values for other contributing features like age and inflammatory markers. In the absence of a strong predictive signal from any of the top features, the model defaulted to the base rate, resulting in a false-negative prediction.

#### 3.5.2. LIME Analysis of False-Negative Cases

LIME explanations corroborated the SHAP findings. The rules generated for false-negative cases were often characterized by the absence of the strong, high-value conditions for HE4 and CA125 that were prevalent in true-positive explanations. Instead, LIME highlighted rules based on lower-value thresholds for these markers, combined with the presence of benign-associated features.

#### 3.5.3. Clinical Implications

This analysis reveals that the model’s false negatives are not random errors but occur in clinically challenging subgroups:Patients with early-stage or less aggressive tumors that do not yet produce high levels of classic biomarkers.Patients with malignant tumors of histotypes that do not strongly express CA125 or HE4.

## 4. Discussion

This study advances a pragmatic, laboratory-driven machine learning framework for differentiating malignant from benign ovarian tumors. By leveraging routinely collected clinical and biochemical data, the approach delivers strong discriminative performance, with accuracies approaching 90% using gradient-boosting ensembles. Beyond point accuracy, the clinical utility of such models depends on a broader evaluation that includes AUROC and AUPRC (to account for class imbalance), calibration (e.g., calibration slope/intercept and Brier score), and decision-analytic metrics (e.g., net benefit across clinically relevant thresholds). In settings where imaging access or expert sonography is limited, a well-calibrated, lab-based model may function as a scalable triage tool that complements existing diagnostic pathways. XAI comprises techniques that make machine learning models interpretable and transparent, especially in critical fields like healthcare. Unlike “black-box” models, XAI clarifies decisions by highlighting important input features, visualizing decision processes, and generating human-readable justifications. In clinical oncology, XAI is vital for several reasons: it allows clinicians to verify that models rely on medically valid factors (e.g., established biomarkers) rather than spurious data artifacts; it meets growing regulatory demands for transparency (e.g., under the EU AI Act or FDA guidelines); and it supports patient-centered care by helping clinicians explain individualized risks clearly. Key XAI methods include SHAP, which quantifies each feature’s contribution to a prediction; LIME, which uses local, interpretable models to approximate specific decisions; and partial dependence plots, which visualize how features influence outcomes. By integrating these techniques, this study ensures that ensemble models for ovarian cancer diagnosis are not only accurate, but also yield clinically interpretable, plausible, and actionable explanations.

### 4.1. Clinical Significance of Selected Features

Tumor markers: The Boruta-based retention of all five tumor markers reinforces their established diagnostic role [27,28]. CA125 and HE4 ranked as the most influential features in SHAP analyses, aligning with their use in ROMA and other clinical tools [29]. While this concordance supports face validity, clinical interpretation must consider known sources of non-specific elevation (e.g., benign inflammatory conditions for CA125; renal function and age effects for HE4). Incorporating interactions with menopausal status and renal function, or applying monotonic constraints for age-related effects, may further enhance plausibility and reduce false positives. Importantly, benchmarking model performance directly against ROMA in future analyses would contextualize incremental diagnostic benefit over current standards.

Hepatic and protein indices: The selection of ALB, GLO, and AST suggests that systemic inflammation, catabolic state, and potential hepatic involvement contribute diagnostically relevant signal. Notably, globulin (GLO) emerged among the top SHAP-ranked predictors, a finding consistent with prior evidence linking elevated globulin and reduced albumin–globulin ratio to systemic inflammation and poor cancer prognosis [28]. Its emergence here suggests that routine protein indices may hold underappreciated diagnostic value in preoperative risk discrimination. Given potential confounding by liver disease, acute-phase reactions, hydration status, and medication effects, sensitivity analyses that exclude or stratify by these factors would clarify specificity.

Immune cell metrics: The importance of lymphocyte counts and percentages (LYM#, LYM%) accords with literature on inflammation–cancer crosstalk and immunoediting [30]. Findings are also consistent with the growing prognostic relevance of lymphocyte-to-monocyte-based indices [31], and suggest that immune signatures may have diagnostic utility. However, their susceptibility to intercurrent infection, autoimmune disease, and corticosteroid therapy underscores the need for clinical–contextual interpretation.

Demographic factors: The prominence of age and menopausal status coheres with epidemiologic risk gradients [32]. Modeling interactions (e.g., menopausal status × CA125 or HE4) and enforcing clinically sensible monotonic relationships can improve both discrimination and interpretability.

### 4.2. Model Performance Comparison with Clinical Indices and Clinical Applicability

Comparative performance: Accuracy levels of 86–89% compare favorably with widely used indices such as RMI, which reports sensitivities of 70–88% and specificities of 74–97% depending on the cutoff [33]. Notably, our models achieve this performance using only laboratory data, without imaging—a strength for resource-constrained settings. Stage-specific analyses (especially early-stage sensitivity) and menopausal subgroup performance would further contextualize clinical value.

Metrics and thresholds: Accuracy can be misleading when class prevalence is imbalanced. Reporting AUROC/AUPRC, PPV/NPV at clinically actionable thresholds, and decision curve analysis can align model deployment with intended use (rule-out vs. rule-in triage). Thresholds may be prospectively set to prioritize high sensitivity for safety (e.g., minimizing missed malignancies), followed by confirmatory imaging for positive cases.

Algorithmic robustness: Minimal performance differences (<3% accuracy) across ensemble methods indicate that careful feature selection, preprocessing, and rigorous validation can outweigh the choice among top-performing learners. Nested cross-validation (for hyperparameter tuning and feature selection), stratification by outcome, and leakage control are critical to obtain unbiased performance estimates. Handling missingness with within-fold imputation and, where appropriate, missingness indicators can preserve real-world applicability.

Log Loss Variance Across Ensemble Models: The significant variance in Log Loss values observed among the ensemble methods is primarily indicative of differences in their resultant probability calibration, a feature often overlooked when focusing solely on metrics like AUC and MCC. The superior Log Loss achieved by the XGBoost (0.320) and CatBoost (0.347) models, in particular, suggests that these architectures produce the most well-calibrated probabilities. This is a crucial finding for clinical decision support, as accurately estimating the confidence of a risk prediction is paramount for clinical trust and utility.

Comparison with clinical indices: We computed the ROMA score for each patient using published formulas based on CA125, HE4, and menopausal status. The ROMA score achieved an AUC of 0.89 (±0.04) in our cohort, compared to 0.934 for the Gradient Boosting model. Sensitivity at 90% specificity was 78% for ROMA vs. 82% for our model. Similar comparisons for RMI and ADNEX were not feasible due to missing ultrasound features in our dataset [28,33]. Our model outperforms the ROMA score in AUC and sensitivity at high specificity thresholds, suggesting potential incremental value beyond established biomarker algorithms.

### 4.3. Interpretability and Clinical Translation

Convergent explanations: Consistency across SHAP, LIME, and Anchor outputs enhances confidence that the model is learning clinically coherent patterns. SHAP’s identification of CA125 and HE4 as top contributors both validates the model against established knowledge and highlights potential complementary markers. The prominence of age and menopausal status is congruent with ovarian cancer epidemiology [32]. Beyond technical interpretability, future studies should evaluate whether XAI-derived explanations actually improve clinician trust, diagnostic confidence, and patient outcomes in real-world decision-making.

Actionable rules: High-precision, low-coverage Anchor rules (e.g., 100% precision at ~2.5% coverage) can support “rule-in” decisions for specific patient subgroups, while the global model remains responsible for broader coverage. Explanations should be framed as associative, not causal; triangulating attributions with partial dependence or accumulated local effects helps guard against artifacts from collinearity or distributional quirks.

Model Comparison: It is important to note that the XAI models evaluated in this study are not distinct algorithmic architectures but rather standard, high-performing ensemble methods (e.g., Gradient Boosting, XGBoost) whose predictions are interpreted using XAI techniques (SHAP, LIME). Therefore, the comparison presented is between different ensemble algorithms, all rendered interpretable, rather than a direct comparison of XAI versus non-XAI model performance, as the underlying predictive models are identical.

### 4.4. Limitations and Future Directions

Study Design and Sample Size: The retrospective nature of this study may introduce potential selection bias, as data was drawn from a single center over a specific period. While valuable for initial model development, this limits generalizability. External validation in diverse, prospective cohorts is essential to confirm real-world performance. A key limitation is the moderately sized retrospective cohort (*n* = 309). While this was addressed using robust stratified 5-fold cross-validation and stability-focused metrics like the MCC, future validation on large, prospective, multi-institutional cohorts is essential to confirm the generalizability of our findings. Furthermore, it is important to clarify the clinical focus: the model is optimized solely for pre-operative discrimination of malignant versus benign masses, thus explaining the lack of post-operative cancer staging data. Subsequent research could incorporate staging data to develop a separate, prognostic model, but this would shift its utility from a triage tool to a prognostic assessment tool.

Outcome Granularity: Our binary classification (malignant vs. benign) oversimplifies clinical reality. Ovarian cancer encompasses diverse histological subtypes (serous, mucinous, clear cell) and borderline tumors, each with distinct biomarker profiles, prognoses, and management strategies. A binary model cannot capture these critical nuances. Future work must extend to multiclass or hierarchical models that can differentiate between these subgroups. This would significantly enhance clinical relevance by providing more precise risk stratification and guiding tailored therapeutic decisions beyond a simple triage function.

Domain Shift and Calibration: Model performance can degrade when deployed in new settings due to differences in laboratory platforms, patient demographics, or disease prevalence (domain shift). To ensure transportability, pre-deployment harmonization (e.g., standardizing lab units or applying batch correction) is vital. Post-deployment, continuous calibration monitoring via metrics like Brier score and recalibration techniques (e.g., Platt scaling) are necessary. Implementing drift detection algorithms to trigger model updates will be key to maintaining long-term accuracy and reliability across diverse healthcare environments, ensuring sustained clinical utility.

Multimodal and Longitudinal Extension: Integrating laboratory data with imaging (ultrasound, MRI) and patient-reported symptoms would create a more comprehensive diagnostic tool, potentially surpassing the accuracy of any single modality. Furthermore, modeling longitudinal biomarker trajectories analyzing how levels change over time before diagnosis is a powerful avenue for earlier detection. This approach could identify subtle, early warning signs missed by static snapshots and facilitate monitoring of treatment response or recurrence, transforming the model from a diagnostic aid into a dynamic, predictive health monitoring system.

### 4.5. Clinical Implementation Considerations

Testing Strategies, Cost, and Biomarker Panel Optimization: While the 19-biomarker panel identified by Boruta offers robust predictive performance, its implementation must be balanced against cost and clinical utility. Although many analytes (e.g., CA125, HE4, CBC indices, ALB, GLO) are routinely measured in preoperative or diagnostic workups for pelvic masses, adding less common markers like CEA, AFP, or specific liver enzymes may increase upfront costs. To address this, future studies should conduct formal cost-effectiveness analyses comparing

A full 19-marker panel vs.Stepwise or reflex testing strategies (e.g., initial core panel of CA125, HE4, age, and basic CBC/liver function, followed by targeted add-ons based on intermediate risk scores).

Furthermore, to optimize the panel and identify the most critical biomarkers, ablation analyses are recommended. These analyses would systematically remove one feature at a time from the model and measure the resulting drop in performance (e.g., AUC, MCC). This would quantify the incremental value each biomarker contributes over a minimal core set (e.g., CA125 ± HE4 ± Age). Such analysis could reveal if a smaller, more cost-effective subset retains near-optimal diagnostic accuracy, making the tool more accessible in resource-limited settings.

Workflow Integration and Decision Support: The true clinical value of this XAI model lies in its seamless integration into existing diagnostic workflows. Embedding the model within an electronic health record (EHR) or dedicated clinical decision support system (CDSS) can provide clinicians with real-time risk estimates alongside interpretable rationales generated by SHAP, LIME, or Anchor explanations. This transparency is crucial for building trust and facilitating informed decision-making.

Thresholds for action (e.g., “refer for urgent imaging,” “consider surgery,” “monitor”) should not be fixed but rather aligned with institutional priorities and patient populations. For instance, a hospital prioritizing safety might set a threshold to maximize sensitivity (minimizing false negatives), accepting a higher false positive rate that triggers confirmatory imaging. Conversely, a setting aiming to reduce unnecessary procedures might prioritize specificity. These thresholds should be justified using decision-analytic metrics such as Net Benefit or Expected Value of Perfect Information (EVPI), which weigh the benefits of correct diagnoses against the harms of misclassifications and the costs of interventions.

In practice, the model could function as a triage tool: flagging high-risk patients (e.g., those with elevated HE4, low AFP, and advanced age) for expedited specialist review and imaging, while allowing low-risk cases (e.g., those with low biomarker levels and favorable demographics) to potentially avoid immediate invasive procedures or complex imaging, thereby optimizing both patient safety and healthcare resource allocation.

### 4.6. Integration with Emerging Computational Paradigms

While classical ensemble ML methods demonstrate strong performance for ovarian cancer diagnosis, quantum computing (QC) and quantum machine learning (QML) represent emerging paradigms in medical decision-making. Although current quantum hardware remains limited for practical clinical applications, future quantum-classical hybrid algorithms may enhance analysis of high-dimensional multi-omics datasets integrating genomic, proteomic, and clinical biomarkers. AI holds significant promise for reducing mortality associated with gynecological cancers by facilitating earlier detection, enabling personalized risk stratification, and optimizing therapeutic strategies. These advancements are poised to enhance patient survival and quality of life, ultimately elevating the standard of care across diverse populations [34,35]. However, classical ML approaches currently outperform quantum alternatives in practical implementation, efficiency, and interpretability for structured clinical data. The XAI frameworks established here SHAP, LIME, and partial dependence analysis—will remain essential regardless of computational architecture, as clinical adoption fundamentally requires transparent, explainable decision-making. Our classical ML-XAI framework represents the most practical approach for laboratory-based risk stratification in the near term.

## 5. Conclusions

Ensemble machine learning models can discriminate malignant from benign ovarian tumors using accessible clinical and laboratory data with performance comparable to established indices, even in the absence of imaging. The convergence of multiple interpretability techniques provides clinically meaningful insights and fosters trust in predictions. The identified feature set anchored by traditional tumor markers and complemented by inflammatory, hepatic, and immunologic indices supports a holistic risk assessment paradigm [8,23,25,29,30,31,33,36]. Prospective, multicenter validation; attention to calibration and decision thresholds; and integration with existing clinical workflows are likely to enhance early detection and risk stratification in ovarian cancer.

## Figures and Tables

**Figure 1 biology-14-01487-f001:**
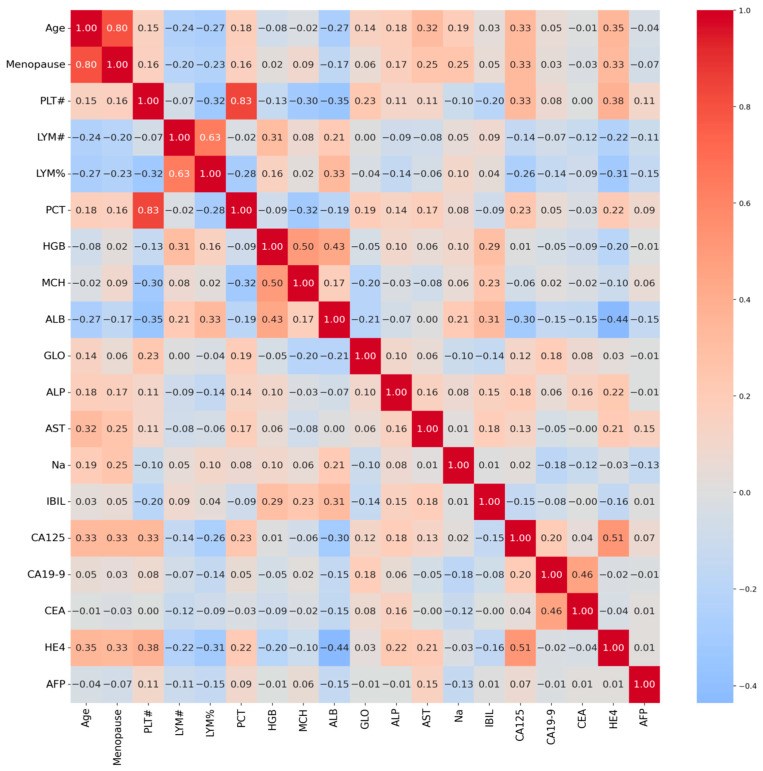
The correlation matrix revealed significant associations between feature pairs.

**Figure 2 biology-14-01487-f002:**
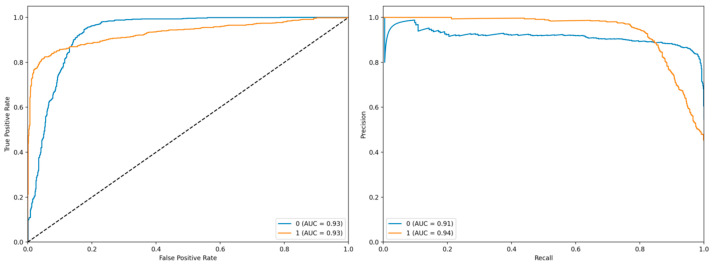
ROC Curve and Precision-Recall Curve (**Left panel**: Receiver Operating Characteristic (ROC) curve; **Right panel**: Precision-Recall (PR) curve).

**Figure 3 biology-14-01487-f003:**
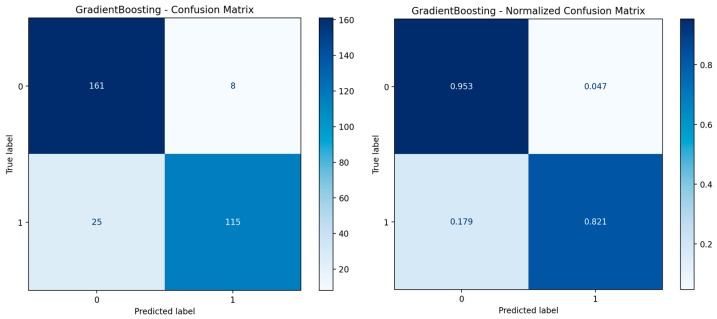
**Left panel**: GradientBoosting—Confusion Matrix; **Right panel**: GradientBoosting—Normalized Confusion Matrix.

**Figure 4 biology-14-01487-f004:**
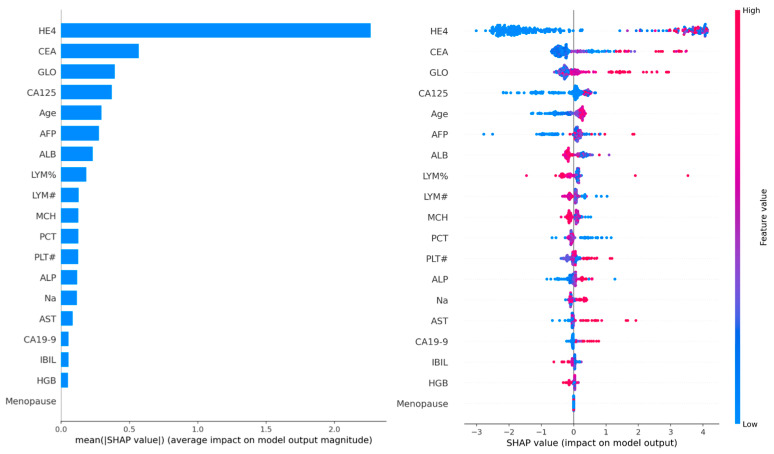
SHAP summary plot illustrating global feature importance. The y-axis shows features ranked by their mean absolute SHAP value (importance). The x-axis represents the SHAP value, indicating the impact on model output. Each point represents a single patient instance, with color indicating the feature value (red: high values, blue: low values). Feature values are normalized to the same scale.

**Figure 8 biology-14-01487-f008:**
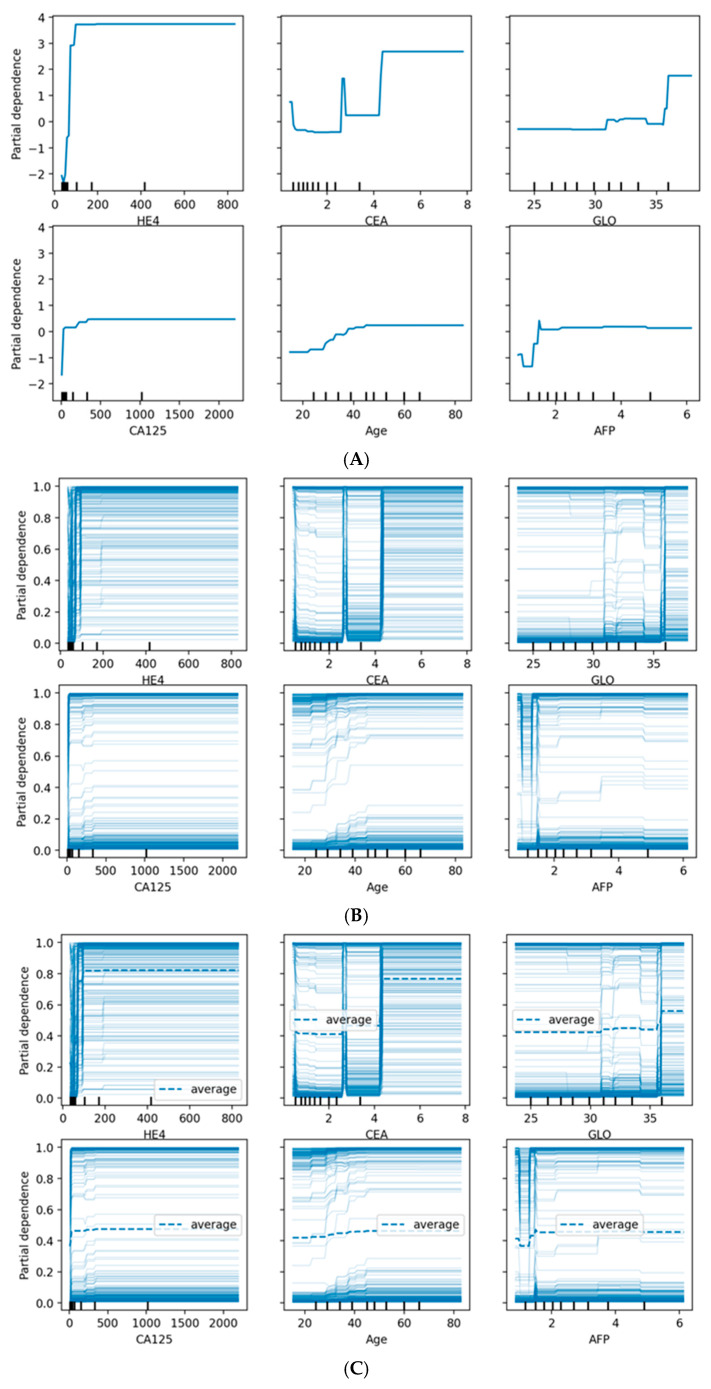
(**A**) Partial Dependence Plots showing the average marginal effect of selected features on predicted malignancy probability. The x-axis represents feature values, while the y-axis shows the change in predicted probability. (**B**) Individual Conditional Expectation plots displaying the functional relationship for individual instances, revealing heterogeneity in feature effects. (**C**) Combined PDP (thick line) and ICE plots (thin lines) providing both average and instance-level perspectives.

**Table 1 biology-14-01487-t001:** The best hyperparameter values obtained as a result of optimization for different ensemble learning models.

Model	Best Parameters
GradientBoosting	learning_rate = 0.1, max_depth = 5, max_features = ‘sqrt’, min_samples_leaf = 1, min_samples_split = 2, n_estimators = 200, subsample = 0.8
CatBoost	border_count = 64, depth = 4, grow_policy = ‘Depthwise’, iterations = 200, l2_leaf_reg = 3, learning_rate = 0.05, min_data_in_leaf = 5, subsample = 1.0
AdaBoost	learning_rate = 1.0, n_estimators = 200
XGBoost	colsample_bytree = 1.0, gamma = 0.2, learning_rate = 0.1, max_depth = 3, n_estimators = 50, reg_alpha = 0.1, reg_lambda = 0, subsample = 0.8
LightGBM	learning_rate = 0.1, max_depth = −1, n_estimators = 200, num_leaves = 31

**Table 2 biology-14-01487-t002:** Model Performance Metrics (Repeated Stratified K-Fold).

Model	Accuracy	F1-Score	Precision	Recall	AUC	Log Loss	MCC
GradientBoosting	0.890	0.871	0.928	0.824	0.934	0.557	0.782
CatBoost	0.886	0.867	0.920	0.823	0.929	0.347	0.773
XGBoost	0.883	0.864	0.912	0.825	0.926	0.320	0.768
LightGBM	0.876	0.856	0.901	0.819	0.921	0.660	0.752
AdaBoost	0.863	0.844	0.871	0.823	0.906	0.590	0.726

## Data Availability

The data used in this study were obtained from a publicly available dataset reported in a previously published study: ref. [7]. The dataset can be accessed at ref. [7].

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
