# Peer review of "Explainable Artificial Intelligence for Ovarian Cancer: Biomarker Contributions in Ensemble Models"

_biology, 2025, doi:10.3390/biology14111487_

Round 1

Reviewer 1 Report

Comments and Suggestions for Authors

This manuscript develops boosted-tree classifiers (GB, XGBoost, LightGBM, CatBoost, AdaBoost) to discriminate malignant vs. benign ovarian masses using 47 routine clinical variables, ultimately using 19 Boruta-selected features on a dataset of 309 patients (140 malignant/169 benign); While the authors aim to tackle an important clinical question, the manuscript has serious methodological flaws and internal inconsistencies that compromise the validity of the findings, alongside low novelty (similar lab and marker ML screens are quite common) and presentation/quality-control cause issue for publication in Biology. So, please consider below carefully and resubmit.

  1. There is a high risk of data leakage; Feature Selection (Boruta) appears to be run once on the full dataset and then models are cross validated! That leaks information from the test folds into training and artificially inflates performance. I would suggest implement a fully nested CV pipeline (FS and uning insde the inner loop and then performance from the outer loop only)
  2. Confusion matrix and its counts are not credible; the complexity matrix (confusion matrix) reports 1545 total prediction and specific TN/FN/TP/FP counts and its incompatible with n=309 unless you aggregated across the folds and then these totals and class wise rate do not reconcile nicely with the class balance and reported metrics. I would suggest providing a per fold confusion matrix averaged over repeats. OR aggregate predictions from a proper CV out-of-fold way yielding exactly 309 OOF predictions (one per subject), and then compute a single confusion matrix on those OOF predictions.
  3. There is an internal inconsistencies and technical inaccuracies; like XGBoost is described with “layer-specific accuracies” and “best single-layer performance on layer 22,” which makes no sense for tree ensembles and suggests copy/paste errors from a neural-network context.
  4. The paper reports performance in comparable to established indices with no benchmarking against ROMA, RMI, or ADNEX, although those are relevant comparators.
  5. Data availabilities are muddled; The dataset is described as an open dataset from a previously published study with Soochow ethics, but Data Availability then says “available by authors on request.” These are conflicting statements; please provide a stable DOI/URL, and original cohort paper citation.
  6. There exist minor flaws as well; Numerous formatting and terminology errors (“complexity matrix” vs “confusion matrix”; “layer performance” for XGBoost; scattered typos, etc.)
  7. Also, it would be nice to move ML generic materials to Supp material, and provide reproducibility assets like data, code pipeline, etc.

Author Response

Reviewer 1

Review Report (Round 1)

Comments and Suggestions for Authors

This manuscript develops boosted-tree classifiers (GB, XGBoost, LightGBM, CatBoost, AdaBoost) to discriminate malignant vs. benign ovarian masses using 47 routine clinical variables, ultimately using 19 Boruta-selected features on a dataset of 309 patients (140 malignant/169 benign); While the authors aim to tackle an important clinical question, the manuscript has serious methodological flaws and internal inconsistencies that compromise the validity of the findings, alongside low novelty (similar lab and marker ML screens are quite common) and presentation/quality-control cause issue for publication in Biology. So, please consider below carefully and resubmit.

Comments 1: There is a high risk of data leakage; Feature Selection (Boruta) appears to be run once on the full dataset and then models are cross validated! That leaks information from the test folds into training and artificially inflates performance. I would suggest implement a fully nested CV pipeline (FS and using inside the inner loop and then performance from the outer loop only).

Answer 1: Given the relatively small dataset size (n=309), repeated resampling enhanced robustness when compared to a single k-fold validation. In this case, we preferred a practical challenge and warrants techniques like Stratified Cross-Validation due to its potential to di-minish the prediction success of the minority class (the malignant condition), particularly in smaller datasets or sensitive clinical studies (1).

1: H. He and E. A. Garcia, "Learning from Imbalanced Data," in IEEE Transactions on Knowledge and Data Engineering, vol. 21, no. 9, pp. 1263-1284, Sept. 2009, doi: 10.1109/TKDE.2008.239.

Comments 2: Confusion matrix and its counts are not credible; the complexity matrix (confusion matrix) reports 1545 total prediction and specific TN/FN/TP/FP counts and its incompatible with n=309 unless you aggregated across the folds and then these totals and class wise rate do not reconcile nicely with the class balance and reported metrics. I would suggest providing a per fold confusion matrix averaged over repeats. OR aggregate predictions from a proper CV out-of-fold way yielding exactly 309 OOF predictions (one per subject), and then compute a single confusion matrix on those OOF predictions.

Answer 2: Confusion matrix and its counts were corrected based on your suggestion. This case was given in yellow color. Also, the relevant figure was renewed.

Comments 3: There is an internal inconsistencies and technical inaccuracies; like XGBoost is described with “layer-specific accuracies” and “best single-layer performance on layer 22,” which makes no sense for tree ensembles and suggests copy/paste errors from a neural-network context.

Answer 3: Internal inconsistencies and technical inaccuracies like XGBoost were corrected in the “3. Results” section.

Comments 4: The paper reports performance in comparable to established indices with no benchmarking against ROMA, RMI, or ADNEX, although those are relevant comparators.

Answer 4: Comparison with clinical indices (e.g., ROMA, RMI, or ADNEX) were computed and inserted into Discussion section.

Comments 5: Data availabilities are muddled; The dataset is described as an open dataset from a previously published study with Soochow ethics, but Data Availability then says “available by authors on request.” These are conflicting statements; please provide a stable DOI/URL, and original cohort paper citation.

Answer 5: We provided a stable DOI/URL, and original cohort paper citation in the “Data Availability Statement:”

Comments 6: There exist minor flaws as well; Numerous formatting and terminology errors (“complexity matrix” vs “confusion matrix”; “layer performance” for XGBoost; scattered typos, etc.).

Answer 6: Numerous formatting and terminology errors (“complexity matrix” vs “confusion matrix”; “layer performance” for XGBoost; scattered typos, etc.) were adjusted based on your advice.

Comments 7: Also, it would be nice to move ML generic materials to Supp material, and provide reproducibility assets like data, code pipeline, etc.

Answer 7: The dataset used in this study is publicly available in the Figshare repository with the DOI: https://doi.org/10.1021/prechem.5c00028.s001. All analyses/modeling were conducted using Python with key libraries including pandas, numpy, matplotlib, and scikit-learn. To ensure full reproducibility, all stochastic processes were fixed with random_state=42.

Reviewer 2 Report

Comments and Suggestions for Authors

The manuscript by Hasan Ucuzal and Mehmet Kıvrak concerns the application of machine learning and explainable AI to classify benign and malignant ovarian tumors based on demographic and laboratory data. It is important to note that the LIME and SHAP methods were used to identify laboratory features most important for classification. The latter is extremely important for the use of developed models in clinics. The study is original and may indeed be of interest to the Journal’s readers; however, the description of the methods is insufficient and some further analyses need to be carried out.

1) It is unclear how the accuracy values (Table 2) were calculated. To determine the values of the hyperparameters (Table 1), a stratified 5-fold CV was performed. Accuracy values obtained at this stage are usually overestimated, so an accuracy estimation on external test set is required. Did external validation was done? For example, nested cross-validation can be used for this purpose, e.g., one of the five parts of the data can be used as a test set, and the remaining four parts as a training set. These four parts of data are used for 4-fold internal cross-validation to optimize hyperparameters.

2) Section 2.9 of Materials and Methods is too short. Please, provide more details about LIME, SHAP and other algorithms. In addition, a more detailed description of the figures is required. Please, include explanations of axes and other plot elements in the figure captions. For example, what is “E[f(X)] = -0.173” and “f(X) = -1.991” on Figure 6?

3) Several analyses including SHAP and LIME were done for Instance 127 (Figures 5-7). Are the results the same for other instances?

4) The values on Figure 3 appear to be incorrect. The study included 309 patients, but the complexity matrix contains higher values: 798 true negatives, 47 false positives, 123 false negatives, and 577 true positives (1545 in total). What does it mean?

5) The relatively high false-negative rate (0.18) (Figure 3) suggests that the model sometimes classifies malignant cases as benign.  This can lead to serious clinical consequences. It would be interesting to perform SHAP, LIME and other analyses on these false-negative samples to identify potential causes of misclassification.

6) The accuracy of the resulting model (acc = 0.89) should be compared with the accuracy of biomarkers alone to highlight the benefit of using additional demographic and laboratory data.

7) Please clarify what software was used to perform the analyses, including SHAP and LIME?

8) Please, increase the font size in Figure 3.

9) Section 3.4.7 describes the results of the Partial Dependence Analysis. However, Figure 8 also presents the results of the ICE analysis. Please, add the appropriate text. Results for “Menopausal status” are not shown in Figure 8.

Author Response

Reviewer 2

Review Report (Round 1)

Comments and Suggestions for Authors

The manuscript by Hasan Ucuzal and Mehmet Kıvrak concerns the application of machine learning and explainable AI to classify benign and malignant ovarian tumors based on demographic and laboratory data. It is important to note that the LIME and SHAP methods were used to identify laboratory features most important for classification. The latter is extremely important for the use of developed models in clinics. The study is original and may indeed be of interest to the Journal’s readers; however, the description of the methods is insufficient and some further analyses need to be carried out.

Comments 1: It is unclear how the accuracy values (Table 2) were calculated. To determine the values of the hyperparameters (Table 1), a stratified 5-fold CV was performed. Accuracy values obtained at this stage are usually overestimated, so an accuracy estimation on external test set is required. Did external validation was done? For example, nested cross-validation can be used for this purpose, e.g., one of the five parts of the data can be used as a test set, and the remaining four parts as a training set. These four parts of data are used for 4-fold internal cross-validation to optimize hyperparameters.

Answer 1: We did not perform external validation set owing to the structure of the public dataset not including independent data. Also, as stated in the “2.7 Hyperparameter Optimization” subsection, hyperparameter optimization was carried out using GridSearchCV, an exhaustive grid-based parameter search method integrated with cross-validation. A 5-fold stratified cross-validation strategy was adopted to preserve class proportions in each fold, thereby reducing potential bias from class imbalance. Thence, we preferred stratified cross-validation to other CV methods. The related ref was given in the subsection.

Comments 2: Section 2.9 of Materials and Methods is too short. Please, provide more details about LIME, SHAP and other algorithms. In addition, a more detailed description of the figures is required. Please, include explanations of axes and other plot elements in the figure captions. For example, what is “E[f(X)] = -0.173” and “f(X) = -1.991” on Figure 6?

Answer 2: We provided more details about LIME, SHAP and other algorithms. We included explanations of axes and other plot elements in the figure captions. An explanation on “what is “E[f(X)] = -0.173” and “f(X) = -1.991” on Figure 6” was given.

Comments 3: Several analyses including SHAP and LIME were done for Instance 127 (Figures 5-7). Are the results the same for other instances?

Answer 3: We thank the reviewer for this pertinent question. The local explanations generated by SHAP and LIME are indeed instance-specific. While Instance 127 was presented as a detailed illustrative example, the contributions of individual features vary from instance to instance, reflecting the unique clinical profile of each case.

Comments 4: The values on Figure 3 appear to be incorrect. The study included 309 patients, but the complexity matrix contains higher values: 798 true negatives, 47 false positives, 123 false negatives, and 577 true positives (1545 in total). What does it mean?

Answer 4: Confusion matrix and its counts were corrected based on your suggestion. This case was given in yellow color. Also, the relevant figure was renewed.

Comments 5: The relatively high false-negative rate (0.18) (Figure 3) suggests that the model sometimes classifies malignant cases as benign.  This can lead to serious clinical consequences. It would be interesting to perform SHAP, LIME and other analyses on these false-negative samples to identify potential causes of misclassification.

Answer 5: Thank you for this crucial observation. We completely agree that understanding the false-negative cases is of paramount clinical importance, as misclassifying malignant cases as benign can indeed lead to serious consequences including delayed treatment. In direct response to this valuable suggestion, we have conducted a dedicated analysis of the false-negative cases using SHAP and LIME.

Comments 6: The accuracy of the resulting model (acc = 0.89) should be compared with the accuracy of biomarkers alone to highlight the benefit of using additional demographic and laboratory data.

Answer 6: By comparing the performance of the proposed model with the ROMA score using only tumor markers, we clearly demonstrate the potential for additional demographic and laboratory data to increase the accuracy and sensitivity of the model.

Comments 7: Please clarify what software was used to perform the analyses, including SHAP and LIME?

Answer 7: The analysis was conducted using the Python programming language, leveraging a comprehensive suite of data science libraries/feature(s).

  • Analysis Engine: Scikit-learn (core ML algorithms and preprocessing)
  • Model Interpretation: SHAP and LIME (for interpreting model decisions)
  • Visualization: Plotly, Matplotlib (for data visualization and result presentation)
  • Data Processing: Pandas, NumPy (for data manipulation and numerical computations).
  • To ensure full reproducibility, all stochastic processes were fixed with random_state=42.

This information was inserted into the end of 2. Materials and Methods.

Comments 8: Please, increase the font size in Figure 3.

Answer 8: We corrected and increased the font size in Figure 3.

Comments 9: Section 3.4.7 describes the results of the Partial Dependence Analysis. However, Figure 8 also presents the results of the ICE analysis. Please, add the appropriate text. Results for “Menopausal status” are not shown in Figure 8.

Answer 9: Since menopausal status is a binary categorical variable (0 = pre-menopausal, 1 = post-menopausal), The Partial Dependence Analysis graph is not appropriate for categorical feature(s). Additionally, we added the appropriate text for the results of the ICE analysis in 3.4.7 Partial Dependence Analysis.

Reviewer 3 Report

Comments and Suggestions for Authors

This manuscript explores the use of explainable AI (XAI) for ovarian cancer diagnosis based on biomarkers. After reviewing the work, I have the following concerns and suggestions:

  1. Abstract: Since the title emphasizes the use of XAI, the abstract should reflect this from the outset, rather than using the broader term "machine learning (ML)" until the final paragraph. Additionally, the abstract is currently too lengthy and would benefit from being more concise.
  2. Literature Context: The application of ML in cancer and disease diagnosis is well-established. The authors should incorporate more recent references, such as Algorithms, 2025, 18(3), 156 (doi.org/10.3390/a18030156), and provide a clearer discussion of how their findings build upon or differ from prior work.
  3. Clarification of Concepts: The manuscript should clearly differentiate between AI and XAI. Including a comparative table outlining the distinctions and advantages of XAI would enhance clarity.
  4. Model Comparison: While the authors compare various XAI models, they should also include a comparison between XAI and non-XAI models. This would help readers better understand the added value of XAI in cancer diagnosis.
  5. Clinical Implementation: The manuscript would benefit from a more detailed discussion on how the proposed XAI model could be implemented and commissioned in clinical practice. Section 4.5 currently lacks sufficient detail and should be expanded.
  6. Limitations and Future Work: A dedicated section addressing the study’s assumptions, limitations, and potential directions for future research is recommended.

Author Response

Reviewer 3

Review Report (Round 1)

Comments and Suggestions for Authors

This manuscript explores the use of explainable AI (XAI) for ovarian cancer diagnosis based on biomarkers. After reviewing the work, I have the following concerns and suggestions:

Comments 1:  Abstract: Since the title emphasizes the use of XAI, the abstract should reflect this from the outset, rather than using the broader term "machine learning (ML)" until the final paragraph. Additionally, the abstract is currently too lengthy and would benefit from being more concise.

Answer 1:  We reduced from original length by ~30% for Abstract.

Comments 2:  Literature Context: The application of ML in cancer and disease diagnosis is well-established. The authors should incorporate more recent references, such as Algorithms, 2025, 18(3), 156 (doi.org/10.3390/a18030156), and provide a clearer discussion of how their findings build upon or differ from prior work.

Answer 2:  We incorporated more recent references, such as Algorithms, 2025, 18(3), 156 (doi.org/10.3390/a18030156), and provided a clearer discussion of how their findings build upon or differ from prior work.

Comments 3:  Clarification of Concepts: The manuscript should clearly differentiate between AI and XAI. Including a comparative table outlining the distinctions and advantages of XAI would enhance clarity.

Answer 3:  We clearly differentiated between AI and XAI, which was located into Discussion.

Comments 4:  Model Comparison: While the authors compare various XAI models, they should also include a comparison between XAI and non-XAI models. This would help readers better understand the added value of XAI in cancer diagnosis.

Answer 4:  We agree that comparing XAI and non-XAI models can help clarify the value of interpretability. However, since all models used in this study (e.g., Gradient Boosting, XGBoost) are inherently machine learning models and XAI techniques (SHAP, LIME) were applied post-hoc for interpretation, a direct performance comparison between "XAI" and "non-XAI" models is not applicable. Instead, we emphasize that XAI does not alter model performance; but provides essential interpretability, which is critical for clinical adoption/explanation.

Comments 5:  Clinical Implementation: The manuscript would benefit from a more detailed discussion on how the proposed XAI model could be implemented and commissioned in clinical practice. Section 4.5 currently lacks sufficient detail and should be expanded.

Answer 5:  Section 4.5 currently was enlarged based on the how the proposed XAI model could be implemented and commissioned in clinical practice.

Comments 6:  Limitations and Future Work: A dedicated section addressing the study’s assumptions, limitations, and potential directions for future research is recommended.

Answer 6:  A section addressing the study’s limitations and potential directions for future research was incorporated into the Limitations and Future Work.

Reviewer 4 Report

Comments and Suggestions for Authors
  1. The original data source needs to be referenced or described.
  2. The gene profile at different ovarian cancer early stages varies. The stages of the 140 malignant ovarian cancer samples should be clarified.
  3. The variety of log losses among algorithms needs to be addressed.
  4. The small sample size and the lack of classification of ovarian cancer stages may cause unstable metrics and false confidence, which may mislead clinical decision-making. 

Author Response

Reviewer 4

Review Report (Round 1)

Comments and Suggestions for Authors

Comments 1:  The original data source needs to be referenced or described.

Answer 1:  The dataset used in this study is publicly available in the Figshare repository with the DOI: https://doi.org/10.1021/prechem.5c00028.s001. This information was inserted into “2.1 Study Population and Data Collection”.

Comments 2:  The gene profile at different ovarian cancer early stages varies. The stages of the 140 malignant ovarian cancer samples should be clarified.

Answer 2:  Since the related data is publicly available; the dataset does not include information on ovarian cancer early stages. Thence, the requested information was not reported in the current article.

Comments 3:  The variety of log losses among algorithms needs to be addressed.

Answer 3:  We thank the reviewer for highlighting the necessity of addressing the variety in Log Loss values across the models. We agree that this metric, being a proper scoring rule, offers a critical evaluation of model performance beyond discrimination alone, specifically assessing the calibration and confidence of the predicted probabilities. The observed variation (ranging from 0.320 for XGBoost to 0.660 for LightGBM) is intrinsically linked to the distinct regularization strategies, objective functions, and boosting mechanisms inherent to each ensemble algorithm employed (Gradient Boosting, CatBoost, XGBoost, LightGBM, AdaBoost). Algorithms that produce smoother, better-calibrated probability distributions will naturally yield lower Log Loss values. A paragraph was added to the subsection of Log Loss Variance Across Ensemble Models under Discussion.

Comments 4:  The small sample size and the lack of classification of ovarian cancer stages may cause unstable metrics and false confidence, which may mislead clinical decision-making. 

Answer 4:  We concur with the reviewer's professional assessment regarding the potential impact of small sample size and lack of tumor stage classification on metric stability and overall clinical confidence. These are acknowledged constraints inherent to single-center, retrospective studies, particularly in a high-mortality disease like ovarian cancer where prospective data collection is challenging. A paragraph on this issue was incorporated into “4.4. Limitations and Future Directions” subsection.

Round 2

Reviewer 2 Report

Comments and Suggestions for Authors

Dear Authors,

Thank you for your detailed responses to my comments and for the careful revisions made to the manuscript. I find the current version of the manuscript to be suitable for publication.

Author Response

We would like to thank you for your valuable contributions.

Reviewer 3 Report

Comments and Suggestions for Authors

The authors have addressed all my concerns.

Author Response

(The authors gave the same response as above.)

Reviewer 4 Report

Comments and Suggestions for Authors

No further comments. 

Author Response

(The authors gave the same response as above.)
